# Cardiopulmonary resuscitation in veno-venous-ECMO patients—A retrospective study on incidence, causes and outcome

Hendrik Booke[1,2], Kai Zacharowski[1], Elisabeth Hannah Adam[1], Florian Jürgen Raimann[1], Frederike Bauer[1], Armin Niklas Flinspach[1] *

1 Department of Anaesthesiology, Intensive Care Medicine and Pain Therapy, University Hospital Frankfurt, Goethe-University Frankfurt, Frankfurt/Main, Germany, 2 Department of Anesthesiology, Intensive Care and Pain Medicine, University Hospital Muenster, University of Muenster, Muenster, Germany

* armin.flinspach@kgu.de

## Abstract

### Introduction

Cardiac arrest in a modern intensive care unit (ICU) is associated with poor outcome although optimal resources are present at all times. Data on cardiac arrest (CA) of the increasing cohort of patients with veno-venous-extracorporeal membrane oxygenation (VV-ECMO) are not available. Due to the highly invasive nature of this procedure, other incidences and causes of cardiac arrest are expected when compared to the ICU population without ECMO. This study focuses on cardiac arrest under VV-ECMO treatment.

### Methods

Retrospective single-center observational study including all VV-ECMO patients from 1st January 2019 until 31st March 2022. Primary focus of this study was number and causes for CA during VV-ECMO treatment. Secondary endpoints were treatment procedure, complications and outcome.

### Results

140 patients were treated with VV-ECMO in the study period. Of those, 23 patients had 29 CA with need for cardiopulmonary resuscitation (CPR) during VV-ECMO treatment. Nearly half of all CA (48%; n = 14) occurred during medical procedures and 21% (n = 6) were device related. Pulseless electric activity (PEA) was the most common rhythm upon CPR initiation (72%). ROSC was achieved in 86%, two CA (6.9%) resulted in extracorporeal CPR. Survival to hospital discharge was 13% following CPR.

### Conclusion

CA occurs in over 15% of all patients treated with a VV-ECMO. Medical procedures during VV-ECMO are associated with a high risk of CA and should be planned with care. Also, the rate of ROSC was very high, only a small number of patients survived the overall VV-ECMO treatment course.

**Data Availability Statement:** The dataset supporting the conclusions of this article contained potentially highly sensitive patient data, such that

publication of the original dataset is prohibited by both the jurisdiction and the Ethics Committee under strict local data protection jurisprudence. Data disclosure may be addressed to the corresponding author (ANF) armin.flinspach@kgu.de or the Data Protection Representative datenschutz@kgu.de upon reasonable request.

**Funding:** The authors received no specific funding for this work.

**Competing interests:** A.N.F. received speaker fees from P.J. Dahlhausen & Co. GmbH, Cologne, Germany, received the Sedana Medical Research Grant 2020 and Thieme Teaching Award 2022. E.H. A. received a research grant of the German Research Foundation (AD 592/1-1) F.J.R received speaker fees from Helios Germany, university hospital Würzburg and Keller Medical GmbH. FJR received financial support by HemoSonics LLC, pharma-consult Petersohn and Boehringer Ingelheim. K.Z. has received honoraria for participation in advisory board meetings for Haemonetics and Vifor and received speaker fees from CSL Behring and GE Healthcare. He is the Principal Investigator of the EU-Horizon 2020 project ENVISION (Intelligent plug-and-play digital tool for real-time surveillance of COVID-19 patients and smart decision-making in Intensive Care Units). F.B. & H.B. declare that there are no conflicts of interest. The other authors declare that there are no conflicts of interest.This does not alter our adherence to PLOS ONE policies on sharing data and materials.

## Introduction

In hospital cardiac arrest (IHCA) is a common scenario with reports of up to 10 cases per 1000 admissions. Most of these cases happen on intensive care units (ICU) and although optimal resources are available, outcome of IHCA rates from 18 to 20% [1–4]. A special cohort within these patients is the one on extracorporeal membrane oxygenation (ECMO). In recent decades and especially in the past couple of years, ECMO has gained a lot of attention [5]. ECMO can be used in two configurations: veno-arterial configuration (VA-ECMO; substituting or supporting native heart-lung function) as well as veno-venous configuration (VV-ECMO; substituting or supporting native lung function). Recently, the use of the latter has increased due to the high number of patients with solely respiratory failure during the worldwide COVID-19 pandemic. While both configurations mainly use centrifugal pumps to drain blood and reinfuse it to the patient, only VA-ECMO can support the heart whereas VV-ECMO is depended on sufficient native heart function. Subsequently, CA on VV-ECMO has similar consequences as CA in patients not on VV-ECMO i.e. loss of blood flow and peripheral oxygen supply.

Due to the high invasive nature of ECMO-treatment and the patients' high dependence on device function, incidence, causes and outcome of CA might be different in comparison to the general ICU population. Additionally, cardiopulmonary resuscitation (CPR) during ECMO treatment may cause unexpected complications in comparison to other patients. ECMO-patients typically have large cannulas in the area where chest compressions take place. This might further increase the pre-existing risk of bleeding which already sums up to 30% [6]. This study aims to investigate the incidence, causes and outcome of CA on VV-ECMO in a university Hospital.

## Materials and methods

### Study design, setting

This was a retrospective monocentric study at the University ARDS/ECMO-center in Frankfurt, Germany and was conducted in accordance to the Declaration of Helsinki [7]. Therapy was conducted by trained intensivists according to current guidelines [5, 8]. The observational period ranged from 1ˢᵗ January 2019 until 31ˢᵗ March 2022.

All patients with VV-ECMO were screened for CA treated with cardiopulmonary resuscitation (CPR)/chest compressions during the time of VV-ECMO treatment. ECMO therapy was performed using a cardiohelp (Getinge AB, Gothenburg, Sweden), rotaflow (Getinge AB) or rotaflow II (Getinge AB) system. All patients required endotracheal intubation, mechanical ventilation and consecutive sedation as required for therapy. The occurrence and diagnosis of ARDS was based on the internationally concerted BERLIN definition [9].

All patients > 18 years treated with CA during VV-ECMO were eligible for inclusion.

We compared demographic data with CA during VV-ECMO to those without CA on VV-ECMO. We further assessed the number of oxygenator changes and positioning maneuvers and calculated the risk for CPR with these procedures.

The study was approved by the local ethics committee of the University of Frankfurt (#2022–738) and registered at clinicaltrials.gov (NCT05342363) at the Registered 14ᵗʰ April 2022. Informed consent for participation was waived for all patients due to the sole retrospective analysis and the data being pseudonymized.

### Data acquisition

All data was sourced from the patients' medical records (PDMS, Metavision 5.4, iMDsoft, Tel Aviv, Israel and ORBIS®, Agfa HealthCare, Bonn, Germany). This included patient's

demographic data, primary diagnosis leading to VV-ECMO and ECMO-cannulation site. ECMO-parameters (blood flow, gas flow), results of blood gas analyses and presence of vaso-pressors 1h prior to CA and during 24h after CA were noted as well as single doses of epineph-rine and amiodarone during CPR. For analysis of CPR, we used the continuous recording of our patients' vital signs and the CA protocols written shortly after CPR wherever present. Number of CPRs per patient, initial cardiac rhythm, number of electric shocks/defibrillations, time to return of spontaneous circulation (ROSC) or time to change in treatment plan (e.g. palliation) and CPR related complications were analyzed. It was further looked into the cause of CA which was then categorized as 'spontaneous', meaning without patient manipulation, 'during medical procedure', e.g. oxygenator change, and 'device associated', e.g. pump failure, by the authors of this manuscript.

## Statistics

We used Microsoft Excel V16.35 (Microsoft Corp., Redmond, WA, USA) for collecting data. Discrete variables are presented as counts (percentage) and continuous variables as means ± standard deviation (SD). For analysis of demographic differences between resusci-tated VV-ECMO patients and those without CA an unpaired student's t-test was used. All cal-culations/analyses were performed with SPSS (IBM Corp., Version 26, Chicago, IL, USA) and GraphPad Prism (GraphPad Software Inc., San Diego, CA, USA).

## Results

### Incidence, demographics and outcome

Out of 6945 patients admitted to our ICU between 1st January 2019 until 31st March 2022, 140 patients received VV-ECMO treatment. The indication for VV-ECMO treatment was based on severe ARDS of different etiologies such as sepsis, trauma, primary pneumonia including COVID-19 (coronavirus disease 2019) and secondary bacterial infections. Due to the common multifactorial nature of severe ARDS, no clear etiological classification was possible in the majority of cases. From 140 patients, 23 patients (16.4%) had at least one CA during VV-ECMO treatment. In total, 29 CA were observed in this cohort (Fig 1). In all patients, 105 oxygenator changes and 487 positioning maneuvers were performed during an overall treat-ment duration with VV-ECMO of 2599 days.

CA-patients had a mean age of 50 ± 10 years, bodyweight of 94.1 ± 19.3 kilogram, BMI of 32.3 ± 6.6 kg/m$^2$ and were mainly male (n = 18; 78.3%). Indication for VV-ECMO treatment was acute respiratory distress syndrome (ARDS) in 21 patients (91.3%). One patient received VV-ECMO due to sepsis with acute pulmonary worsening and one patient due to pulmonary edema. Cannulation sites were both femoral veins (femoral-femoral) or femoral and internal jugular vein (femoral-jugular) in 13 (56.5%) and 10 (43.5%) cases respectively. Mean days on ECMO were 29.8 ± 24.5 days. Three patients (13.0%) survived to hospital discharge. Full patient characteristics are shown in Table 1.

### CPR data

Out of 29 CA in the observational period, 14 (48%) occurred during medical procedures (8 oxygenator changes, 4 positioning maneuvers, 1 shortly after ECMO initiation, 1 tracheal can-nula change) and 6 (21%) were device related (5 pump failures and one battery failure during transport to another room). For the positioning procedures and the tracheal cannula change, all CA protocols state abdominal muscular clenching against the respirator (similar to a Val-salva maneuver) with lost ECMO blood flow and subsequent hypoxemia as the causal chain

# CONSORT

## Mechanical cardiopulmonary resuscitation during extracorporeal membrane oxygenation.

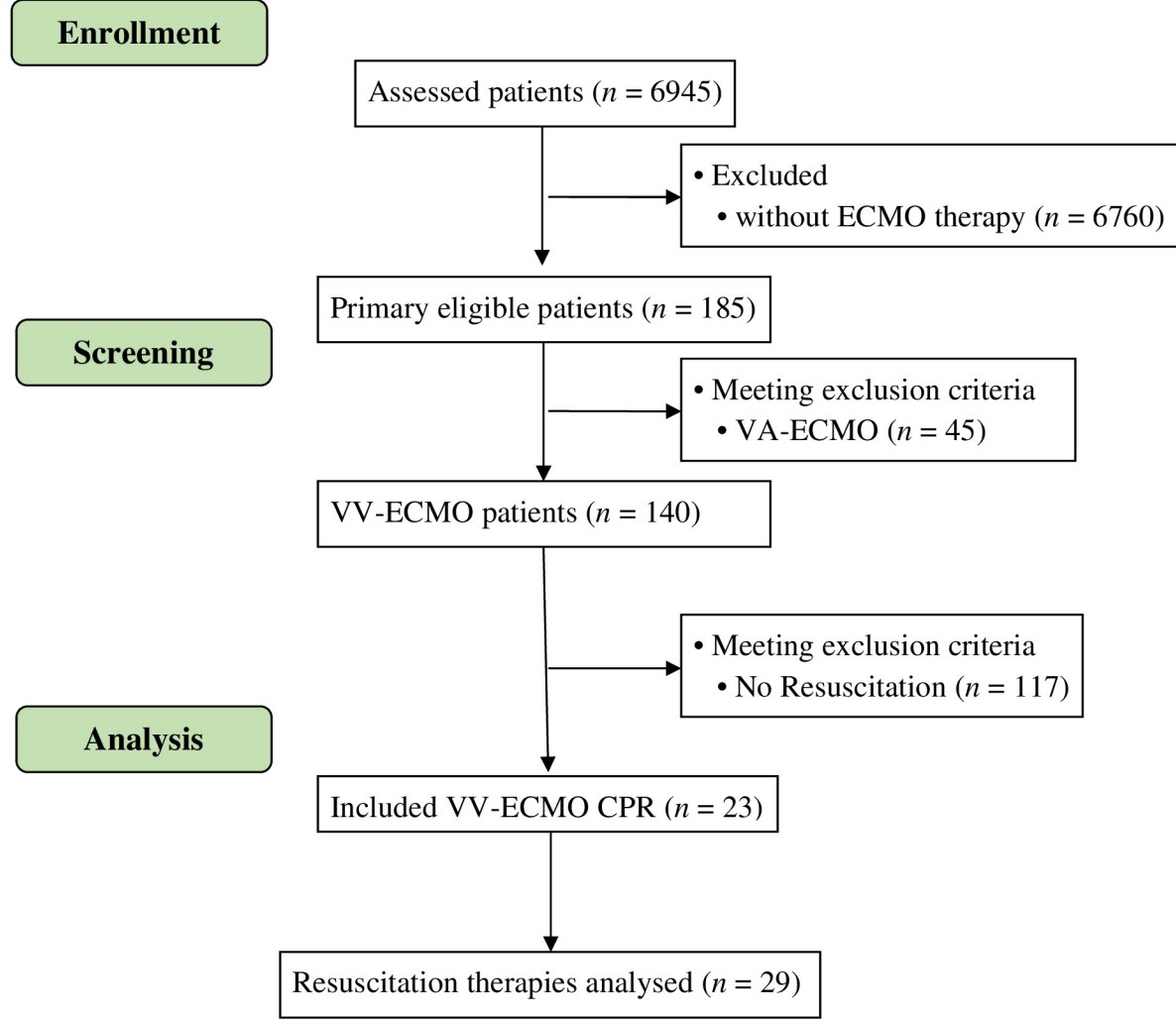

**Fig 1. Study inclusion of patients according to the CONSORT criteria & flow-chart.** Flow chart of patients included into the study with VV-ECMO CPR (n = 23 containing; n = 29 resuscitations) (according to the CONSORT criteria). Abbreviations: CPR, cardiopulmonary resuscitation; VA-ECMO, veno-arterial-extra corporeal membrane oxygenation; VV-ECMO, veno-venous-extra corporeal membrane oxygenation.

for CA. In cases of oxygenator change and device failure the planned or unplanned ECMO support likewise resulted in hypoxemia and CA (Fig 2). Compared to the total number of oxygenator changes (105) and the total number of positioning maneuvers (487), this results in a risk for CA during these procedures of 7.6% (8 of 105) and 0.8% (4 of 487), respectively.

Nine CA (31.0%) were classified as spontaneous, meaning that no intervention took place when CA occurred. In retrospect analysis, two of them were caused by abdominal muscular clenching with subsequent loss of ECMO blood flow and one was linked to hypotension in acute septic shock. The causes for the other five spontaneous occurrences were not found at the time and were neither determinable in retrospect. Four of those unclear CA had ventricular tachycardia (VT), ventricular fibrillation (VF) and one showed no electrical activity (asystole).

**Table 1. Patient characteristics, ECMO configuration, CPR data and outcome data.**

| | n [%] or mean ± SD |
|---|---|
| **Patient characteristics** | |
| Total number patients | 23 |
| Total number CPR | 29 |
| Male | 18 [78] |
| Age [y] | 49.8 ± 10.4 |
| Bodyweight [kg] | 94.1 ± 19.3 |
| BMI [kg/m$^2$] | 32.3 ± 6.6 |
| Diabetes | 8 [35] |
| CVD | 1 [4] |
| CKI | 0 [0] |
| aHT | 9 [39] |
| Smoker | 2 [9] |
| **Diagnosis** | |
| ARDS | 21 [92] |
| Pneumonia | 1 [4] |
| Lung edema | 1 [4] |
| **ECMO characteristics** | |
| Fem-fem cannulation | 13 [57] |
| Fem-jug cannulation | 10 [43] |
| Days on ECMO | 29.8 ± 24.5 |
| **Outcome Parameters** | |
| ICU stay days | 38.2 ± 28.6 |
| Hospital stay days | 40.4 ± 32.5 |
| Survival until discharge | 3 [13] |
| Days from ECMO to CPR | 16.6 ± 18.1 |
| Days from last CPR to Death | 7.4 ± 3.5 |

Table showing the total number of patients, their characteristics and their comorbidities. Abbreviations: aHT: arterial hypertension; ARDS: acute respiratory distress syndrome; BMI: body mass index; CPR: cardiopulmonary resuscitation; CKI: chronic kidney injury; CVD: cardiovascular disease; ECMO: extracorporeal membrane oxygenation; fem-fem: femoral-femoral; fem-jug: femoral-jugular; ICU: intensive care unit; SD: standard deviation; y: years

In total, 21 (72.0%) pulseless electric activities, 5 (17.2%) VT or VF and 3 (10.3%) asystoles were observed as initial rhythms. Electric shocks were administered in 4 CPRs with VT/VF and a total of 10 shocks each 200 Joule biphasic were given (lifepak 20, Medtronic, Dublin, Ireland).

Single doses of adrenalin were given in 21 cases with a mean total dose of 2.0 ± 3.3 mg per CPR. Amiodarone was administered in 2 cases of therapy refractory VT/VF. In 11 cases the continuous application of an additional catecholamine (adrenaline, norepinephrine or argipressin) was necessary within 24h after CA.

ROSC was achieved in 25 out of 29 cases (86.2%) and mean time to ROSC was 3.3 ± 4.4 min. In cases of CA during necessary oxygenator changes, ROSC was achieved in 100% after a mean time of 2.1 ± 1.6 min. In 4 cases (13.8%), ROSC was not achieved, and therapy was changed to palliation (n = 2) or extracorporeal CPR (eCPR) (n = 2). All of those cases were "spontaneous CA" and none survived to hospital discharge. Mean time to death from (last) CPR was 7.4 ± 8.8 days (range 0 to 32 days, median 3.5 days, interquartile range: 14.5 days).

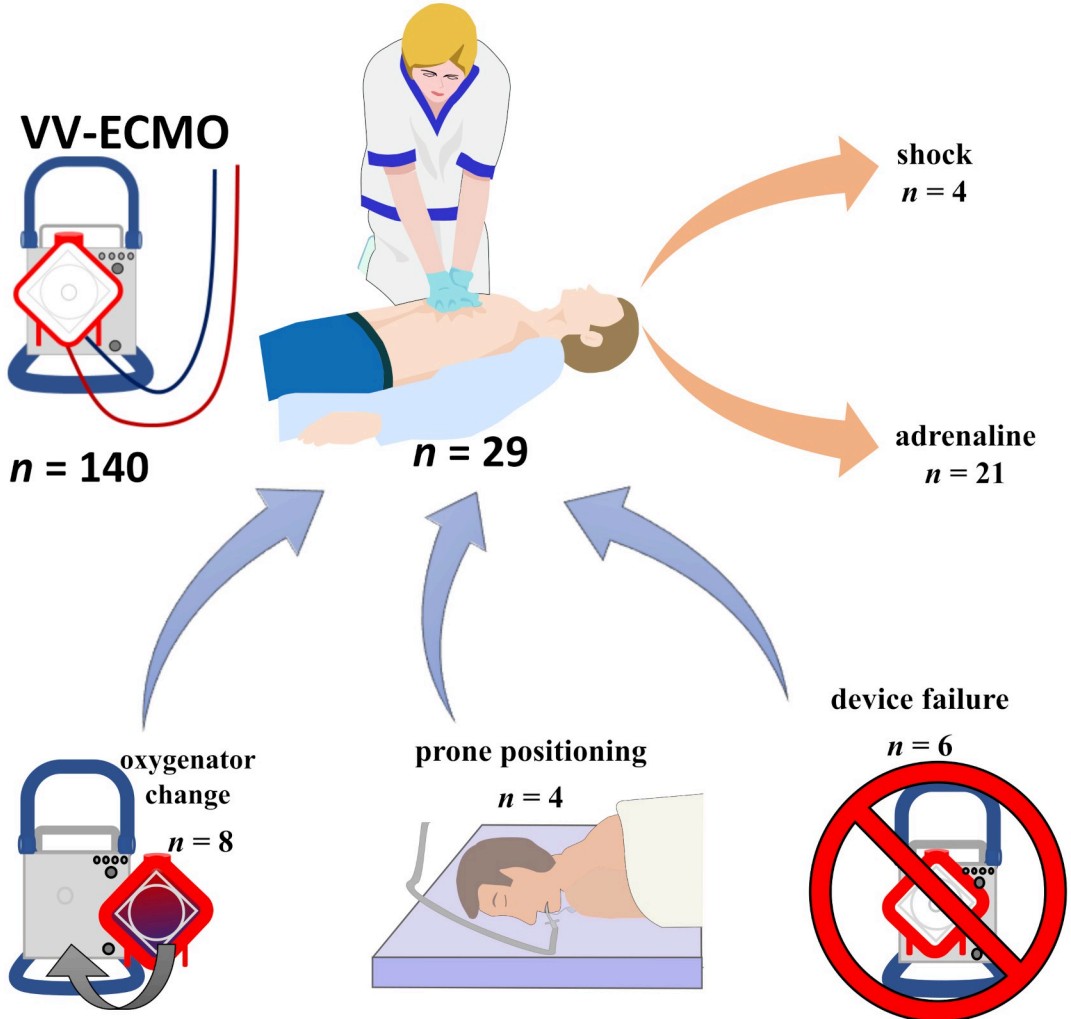

**Fig 2. Cardiopulmonary resuscitation under VV-ECMO.** Reasons for cardiopulmonary resuscitation under existing venovenous ECMO therapy due to severe respiratory failure, and therapeutic approaches according to the AHA guidelines, along with mechanical compression. Abbreviations: VV-ECMO: venovenous extracorporeal membrane oxygenation.

Complications presumably related to the procedure of CPR were observed in 4 patients. One hemopneumothorax, one pneumothorax, one cannula dislocation and one tracheal bleeding. Except for the cannula dislocation, these complications were in patients with femoral-jugular cannulation.

No significant difference was found between the demographic data of the resuscitated VV-ECMO patients and those without CA regarding age, BMI, gender or comorbidities according to medical history. However, there was a significantly increased mortality (p = 0.011) among the patients with CA and consecutive resuscitation under VV-ECMO.

## Discussion

This monocentric, retrospective study on 140 VV-ECMO cases delivers first insights on incidence, causes and outcome of CA during VV-ECMO. We observed a high incidence of nearly 200 CA per 1000 ECMO patients, which is higher than rates reported in the general ICU

cohort [1, 2]. At the same time, the association to medical procedures is high and more than half of CA were directly linked to the ECMO treatment itself (8 oxygenator changes and 6 device failures). Likewise, in case of pump failure or battery failure, the short time of lost ECMO support until manual pumping or restart of the device led to immediate hypoxemia and consecutive CA. In broader perspective, lost ECMO support makes up 65.5% (19 of 29) of all CA we observed. This illustrates the patients' dependence on VV-ECMO function, and also explains our short CPR times as CPR was mostly needed until restart of ECMO support. During planned loss of ECMO flow for oxygenator changes, even shorter CPR times were observed, and uptake of sufficient heart function was seen in 100% after restart of ECMO flow.

In our study, we report four major complications that we classified as CPR related. One hemopneumothorax and one pneumothorax were detected after patient positioning related CA where we cannot determine indefinitely whether CPR was the cause or patients clenching itself.

In our analysis, oxygenator change inherits a 7.6% risk of CA. However, these CA showed 100% ROSC and none of the observed major complications occurred after oxygenator change related CA. This shows that oxygenator changes are a high-risk procedure with increased risk for CA but at the same time a well manageable CA. Nevertheless, the high risk for CA should be considered when weighing risks and benefits of oxygenator change.

In cases of patient positioning leading to CA and the one case during tracheal cannula change, clenching of the patient and suctioning of the drainage tube with subsequent ECMO dysfunction resulted in hypoxia and CA. Increased sedation or the use of neuromuscular blocking agents might have prevented CA here, which is also directly reported in one CA-protocol. Overall, the risk for CA associated to prone positioning therapy seems to be low, with only 0.8% of all positioning maneuvers leading to CA, making prone positioning under ECMO-therapy relatively safe.

The amount of ROSC in this cohort is higher than in previous reports of IHCA [10], but has a low survival to hospital discharge which is the half of other IHCA patients (approximately 18–20%) [1–4]. Bearing in mind the general sickness of VV-ECMO-patients, however, our outcome to hospital discharge is conclusive [8]. Putting together, this shows, that even though CA during VV-ECMO is well manageable with good short-term outcome. It limits the general prognosis substantially.

This study has some limitations caused by its design, which firstly relies on adequate documentation of CPR procedures for adequate retrospective analysis. This also means, that no standardized diagnostics (e.g. sonography-based assessment of ventricular function) took place. For the general purpose of this study, determining incidence, cause and general outcome, however, this has little impact. Secondly, our cohort, based on a monocentric design, is relatively small with only 140 patients included, and our observed incidence makes for the possibility of a much larger investigation in CA during VV-ECMO.

## Conclusions

CA during VV-ECMO treatment is a frequently observed condition with an incidence of over 150 per 1000 ECMO-runs in our cohort. Most CA are linked to medical procedures or the ECMO itself and adequate precautionary measures should take place. Even though, CA during VV-ECMO has high ROSC numbers, it limits the overall prognosis substantially.

## Author Contributions

**Conceptualization:** Hendrik Booke, Armin Niklas Flinspach.

**Data curation:** Hendrik Booke, Frederike Bauer, Armin Niklas Flinspach.

**Formal analysis:** Hendrik Booke, Elisabeth Hannah Adam, Florian Jürgen Raimann, Frederike Bauer, Armin Niklas Flinspach.

**Investigation:** Hendrik Booke, Armin Niklas Flinspach.

**Methodology:** Hendrik Booke, Florian Jürgen Raimann, Armin Niklas Flinspach.

**Project administration:** Kai Zacharowski, Armin Niklas Flinspach.

**Resources:** Kai Zacharowski, Elisabeth Hannah Adam.

**Software:** Hendrik Booke, Armin Niklas Flinspach.

**Supervision:** Kai Zacharowski, Armin Niklas Flinspach.

**Validation:** Hendrik Booke, Armin Niklas Flinspach.

**Visualization:** Hendrik Booke, Armin Niklas Flinspach.

**Writing – original draft:** Hendrik Booke, Elisabeth Hannah Adam, Florian Jürgen Raimann, Frederike Bauer, Armin Niklas Flinspach.

**Writing – review & editing:** Hendrik Booke, Kai Zacharowski, Elisabeth Hannah Adam, Florian Jürgen Raimann, Frederike Bauer, Armin Niklas Flinspach.

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
