## [Decision Letter · Decision Letter 0]

24 Apr 2023

PONE-D-22-31617Cardiopulmonary Resuscitation in veno-venous-ECMO patients – A retrospective study on incidence, causes and outcome.PLOS ONE

Dear Dr. Flinspach,

Thank you for submitting your manuscript to PLOS ONE. After careful consideration, we feel that it has merit but does not fully meet PLOS ONE’s publication criteria as it currently stands. Therefore, we invite you to submit a revised version of the manuscript that addresses the points raised during the review process.

We look forward to receiving your revised manuscript.

Kind regards,

Andrea Ballotta

Academic Editor

PLOS ONE

Journal Requirements:

"A.N.F. received speaker fees from P.J. Dahlhausen & Co. GmbH, Cologne, Germany, received the Sedana Medical Research Grant 2020 and Thieme Teaching Award 2022.

E.H.A. received a research grant of the German Research Foundation (AD 592/1-1)

F.J.R received speaker fees from Helios Germany, university hospital Würzburg and Keller Medical GmbH. FJR received financial support by HemoSonics LLC, pharma-consult Petersohn and Boehringer Ingelheim.

K.Z. has received honoraria for participation in advisory board meetings for Haemonetics and Vifor and received speaker fees from CSL Behring and GE Healthcare. He is the Principal Investigator of the EU-Horizon 2020 project ENVISION (Intelligent plug-and-play digital tool for real-time surveillance of COVID-19 patients and smart decision-making in Intensive Care Units).

F.B. & H.B. declare that there are no conflicts of interest.

The other authors declare that there are no conflicts of interest."

5. We note that Figure 2 in your submission contain copyrighted image. All PLOS content is published under the Creative Commons Attribution License (CC BY 4.0), which means that the manuscript, images, and Supporting Information files will be freely available online, and any third party is permitted to access, download, copy, distribute, and use these materials in any way, even commercially, with proper attribution. For more information, see our copyright guidelines: http://journals.plos.org/plosone/s/licenses-and-copyright.

Additional Editor Comments:

Tx for your contribution. On the basis of the two reviewers' evaluation the paper should undergo to major revision

Reviewers' comments:

Reviewer's Responses to Questions

**Comments to the Author**

1. Is the manuscript technically sound, and do the data support the conclusions?

Reviewer #1: Yes

Reviewer #2: Yes

2. Has the statistical analysis been performed appropriately and rigorously? 

Reviewer #1: Yes

Reviewer #2: I Don't Know

3. Have the authors made all data underlying the findings in their manuscript fully available?

Reviewer #1: Yes

Reviewer #2: No

4. Is the manuscript presented in an intelligible fashion and written in standard English?

Reviewer #1: Yes

Reviewer #2: Yes

5. Review Comments to the Author

Reviewer #1: Thank you for your paper.

It is an interesting work despite is monocentric and retrospective

Please correct at line 56 heart and heart-lung function, is a repetition and amiodaron in amiodarone at line 97

It would be better include a table in which is specified general parameters of the patients and at least diagnosis.

There is no mention of the ventricular function pre and after the event which could be a guidance on the treatment.

When you speak about palliation approach and conversion to eCPR is not clear what was the indication of those.

I have observed a large amount of circuit change (105) for a VV ECMO, you can justified that.

It would be great if you can add the ECMO run time at least a mean and median

Reviewer #2: Dear Author,

I find that the primary purpose of your manuscript is very interesting. However, I have some tips:

- the primary aim of the study is not completely clear, you could better clarify it in the Introduction;

- it would be useful to include soma table to show your data and results;

- what tests did you use for statistical analysis?; You could clarify that in Statistics;

- about anthropometric features, maybe, it would be better to consider Body Mass Index (BMI) than bodyweight;

Furthermore, I think you should better specify the evaluation of criticall ilness that leads to ECMO V-V. For example, you included 21 patients with ARDS, that is just a syndrome. So, what are the causes of ARDS in each patient? What are the criteria you used to define ARDS? Did you include patients with COronaVirus Disease (COVID) in this class of patients or not? If yes, how many patients had COVID and how many patients had a respiratory failure of other etiology? There were differences about spontaneous CA, for example about the incidence of pump failure, between COVID and not-COVID patients?

About the initial rhythms of cardiac arrest (CA), you showed 21 PEA, 5 VT or VF and 3 asystole. It would be interesting to know what rhythm did you observe in spontaneous CA, in order to hypothesize also the underlying causes in the discussion.

Finally, there were differences in CPR related major complications about different cannulation sites?

Best Regards

6. PLOS authors have the option to publish the peer review history of their article (what does this mean?). If published, this will include your full peer review and any attached files.

Reviewer #1: No

Reviewer #2: **Yes: **Marcucci Riccardo

---

## [Author Response · Author response to Decision Letter 0]

25 May 2023

Revision

Journal Requirements:

https://journals.plos.org/PLOSOne_formatting_sample_main_body.pdf

and

https://journals.plos.org/PLOSOne_formatting_sample_title_authors_affiliations.pdf

We revised manuscript accordingly and it now meets PLOS ONE’s style requirements. 

2. Thank you for stating the following in the Competing Interests section: "..."

Our cover letter now includes the statement above in which we declare that our disclosed interests do not alter our adherence to PLOS ONE policies on sharing data and materials.

The original data contains potentially highly sensitive patient data, so that according to the local strict data protection jurisdiction, publication of the original data set is prohibited by the jurisdiction as well as the ethics committee. 

We adjusted our data availability statement accordingly:

“The dataset supporting the conclusions of this article contained potentially highly sensitive patient data, such that publication of the original dataset is prohibited by both the jurisdiction and the Ethics Committee under strict local data protection jurisprudence. Data disclosure may be addressed to the corresponding author (ANF) armin.flinspach@kgu.de or the Data Protection Representative datenschutz@kgu.de upon reasonable request.”

We deleted the ethics declaration at the bottom of our manuscript. It is only found in the methods section now.

5. We note that Figure 2 in your submission contain copyrighted image. All PLOS content is published under the Creative Commons Attribution License (CC BY 4.0), which means that the manuscript, images, and Supporting Information files will be freely available online, and any third party is permitted to access, download, copy, distribute, and use these materials in any way, even commercially, with proper attribution. For more information, see our copyright guidelines: http://journals.plos.org/plosone/s/licenses-and-copyright. 

We would like to apologize to the Editorial Office for the confusion. The graphic was created independently by the Corresponding Author itself using the freely available vector graphics program Inkscape version 1.2.2. Accordingly, the copyright is the exclusive property of the author. If the publisher has any concerns about similarities between the image and other published graphics, please let us know so that we can make appropriate adjustments to our graphics. 

Reviewers' comments:

Reviewer #1:

Thank you for your paper.

It is an interesting work despite is monocentric and retrospective

Please correct at line 56 heart and heart-lung function, is a repetition and amiodaron in amiodarone at line 97

We thank the reviewer for the careful review of the manuscript and have implemented the associated suggestions in the revision.

It would be better include a table in which is specified general parameters of the patients and at least diagnosis.

Thank you very much for your suggestion of including a table describing the patients characteristics. We added a suitable table for further visualization of our patients’ characteristics. 

There is no mention of the ventricular function pre and after the event which could be a guidance on the treatment.

We appreciate the valuable objection raised by the reviewer regarding the assessment of the cardiac function.

Unfortunately, ventricular function was assessed only in a small fraction of patients with regard to the event. Usually, it was not possible to perform a contemporary dedicated cardiac ejection assessment, especially in the context of emergency treatment and resuscitation. The bedside sonographic assessment was in most cases only performed by eyeballing and was not recorded in the patient data management system. Accordingly, no adequate qualitative assessment of the ejection fraction was performed before the event or in the immediate post-event period.

In order to adequately reflect the aspect revealed by the reviewer, we have adjusted the limitations accordingly and now explicitly point out this restriction:

“This also means, that no standardized diagnostics (e.g. sonography-based assessment of ventricular function) took place.”

When you speak about palliation approach and conversion to eCPR is not clear what was the indication of those.

Thank you very much for mentioning this difficult topic of eCPR and palliation of CPR. As there are no clear guidelines for the use of eCPR in intrahospital cardiac arrest and even less so for IHCA in patients on existing VV-ECMO support, our hospital has no standardized operating procedure for the conversion of CPR to eCPR in these patients. This means, that the implementation of an arterial cannula for eCPR was based on the discretion of the attending medical team and a case by case decision. 

Similarly, the decision to end CPR and change the therapy approach towards palliation was made by the attending medical personnel and based on multiple factors (e.g. long CPR time, poor prognosis regarding the overall situation).

If the reviewer likes further clarification, we are happy to further elucidate on this difficult topic. However, we think that the discussion of termination of CPR or change to eCPR is highly complex and does not change the key findings of our work.

I have observed a large amount of circuit change (105) for a VV ECMO, you can justified that.

We thank the reviewer for an attentive observation. These circumstances are due to the relatively long median duration of the ECMO device run of 22 days with an associated cumulative ECMO duration of 2599 days. In this respect, the high number of oxygenator changes was generated from the manufacturer-related maximum duration and the clinically recorded wear parameters, such as pressure drop across the oxygenator, decarboxylation and oxygenation capacity, as well as signs of hemolysis. Consequently, an oxygenator change took place in mean after more than three weeks of total runtime.

In order to better explain this misleading aspect, we have made a corresponding addition to our manuscript.

“In all patients, 105 oxygenator changes and 487 positioning maneuvers were performed during an overall treatment duration with VV-ECMO of 2599 days.”

It would be great if you can add the ECMO run time at least a mean and median

The ECMO run time is now shown in the added table of patient characteristics.

 

Reviewer #2:

Dear Author,

I find that the primary purpose of your manuscript is very interesting. However, I have some tips:

- the primary aim of the study is not completely clear, you could better clarify it in the Introduction;

- it would be useful to include soma table to show your data and results;

- what tests did you use for statistical analysis?; You could clarify that in Statistics;

- about anthropometric features, maybe, it would be better to consider Body Mass Index (BMI) than bodyweight;

Dear reviewer,

Thank you very much for your thorough review of our manuscript.

Regarding your first remark, we now included a table with all patient characteristics. We also added the BMI (body mass index) of our patients for better comparison. Concerning the primary aim of our study, this is a descriptive study which aims to look at the incidence and causes of cardiac arrest in veno-venous ECMO-patients as those are currently not described in the literature. We changed our introduction to clarify our study goals. We also broadened the description of our statistical analysis.

For your other suggestions, please see our point by point responses below.

Again, we really appreciate that you offered your valuable time for reviewing our manuscript and believe that our study profits from your inputs. 

Sincerely

The authors

Furthermore, I think you should better specify the evaluation of criticall ilness that leads to ECMO V-V. For example, you included 21 patients with ARDS, that is just a syndrome. So, what are the causes of ARDS in each patient? What are the criteria you used to define ARDS?

We excuse the inadequate description of the ARDS definition. In our study, we refer to the currently valid ARDS definition according to the BERLIN criteria.

We have made a corresponding addition in the methods section of the manuscript:

“The occurrence and diagnosis of ARDS was based on the internationally concerted BERLIN definition.(10)”

Did you include patients with COronaVirus Disease (COVID) in this class of patients or not?

If yes, how many patients had COVID and how many patients had a respiratory failure of other etiology?

We appreciate the reviewer's objection. Due to the exploratory nature of the study, no selection was made according to the etiology of ARDS. Because of the study period being amidst the pandemic, patients with existing primary COVID-19 ARDS were also included. While many patients were diagnosed with COVID-19 in the course of their hospital stay, a clear separation does not seem possible due to the frequently seen secondary bacterial superinfection with deterioration of condition under broad-spectrum antibiotic therapy.

However, a significant number of patients (not clearly quantifiable) developed an indication for VV-ECMO treatment following an already borderline decrease in COVID-19 cycling time (>30) but with clear laboratory evidence of pulmonary bacterial superinfection (especially procalcitonin levels) leading to ARDS diagnosis and consecutive sample collection with broad antibiotic therapy.

We are very pleased to comply with the reviewer’s suggestion by providing a corresponding extension of the reporting requirements and disclosing these facts more widely:

“The indication for VV-ECMO treatment was based on severe ARDS of different etiologies such as sepsis, trauma, primary pneumonia including COVID-19 (coronavirus disease 2019) and secondary bacterial infections. Due to the common multifactorial nature of severe ARDS, no clear etiological classification was possible in the majority of cases.”

There were differences about spontaneous CA, for example about the incidence of pump failure, between COVID and not-COVID patients?

As previously stated, the authors regret that a subgroup analysis between the primary COVID-19 ARDS patients was not possible. This is due to the inability to clearly identify primary COVID-19 ARDS in the heterogeneous overall group.

About the initial rhythms of cardiac arrest (CA), you showed 21 PEA, 5 VT or VF and 3 asystole. It would be interesting to know what rhythm did you observe in spontaneous CA, in order to hypothesize also the underlying causes in the discussion.

Thank you for the suggestion to analyze the initial rhythm in spontaneous cardiac arrest separately. We now included such analysis in our results paragraph.

“Four of those unclear CA had ventricular tachycardia (VT), ventricular fibrillation (VF) and one showed no electrical activity (asystole).”

Finally, there were differences in CPR related major complications about different cannulation sites?

Three major complications were observed in fem-jug cannulation and one major complication in fem-fem cannulation. We added this to our results.

“Except for the cannula dislocation, these complications were in patients with femoral-jugular cannulation.”

1. Force* TADT. Acute Respiratory Distress Syndrome: The Berlin Definition. JAMA. 2012;307(23):2526-33.

---

## [Editor Report · Decision Letter 1]

29 Jun 2023

PONE-D-22-31617R1Cardiopulmonary Resuscitation in veno-venous-ECMO patients – A retrospective study on incidence, causes and outcome.PLOS ONE

Dear Dr. Flinspach,

Thank you for submitting your manuscript to PLOS ONE. After careful consideration, we feel that it has merit but does not fully meet PLOS ONE’s publication criteria as it currently stands. Therefore, we invite you to submit a revised version of the manuscript that addresses the points raised during the review process.

We look forward to receiving your revised manuscript.

Kind regards,

Andrea Ballotta

Academic Editor

PLOS ONE

Journal Requirements:

Additional Editor Comments :

Let's submit your responses two reviewers in order to reach a decision

---

## [Author Response · Author response to Decision Letter 1]

10 Jul 2023

Journal Requirements:

We would like to thank for this valuable advice and have complied with the request for a critical review of the references we made.

We could not identify any references that have been retracted. Due to the considerable time latency between the original publication and the revision of the Declaration of Helsinki that has taken place in the meantime, we have replaced the corresponding original reference by the most recent amendment.

Thus, there has been a replacement of:

PP R. Human Experimentation Code of Ethics of the World Medical Association. Br Med J. 1964;2(5402):177.

through

World Medical A. World Medical Association Declaration of Helsinki: ethical principles for medical research involving human subjects. JAMA. 2013;310(20):2191-4. doi: https://doi.org/10.1001/jama.2013.281053. PubMed PMID: 24141714.

We hope that we have been able to meet the Journal requirements with the comprehensive critical revision of our references and appreciate the critical revision of our manuscript.

---

## [Editor Report · Decision Letter 2]

2 Aug 2023

Cardiopulmonary Resuscitation in veno-venous-ECMO patients – A retrospective study on incidence, causes and outcome.

PONE-D-22-31617R2

Dear Dr. Flinspach,

We’re pleased to inform you that your manuscript has been judged scientifically suitable for publication and will be formally accepted for publication once it meets all outstanding technical requirements.

Kind regards,

Andrea Ballotta

Academic Editor

PLOS ONE

Additional Editor Comments (optional):

My congrats for the acceptance of the manuscript that i deem ready for publication.
---

## [Editor Report · Acceptance letter]

4 Aug 2023

PONE-D-22-31617R2 

Cardiopulmonary Resuscitation in Veno-Venous-ECMO Patients – A retrospective Study on Incidence, Causes and Outcome 

Dear Dr. Flinspach:

I'm pleased to inform you that your manuscript has been deemed suitable for publication in PLOS ONE. Congratulations! Your manuscript is now with our production department. 

Kind regards, 

on behalf of

Dr. Andrea Ballotta 

Academic Editor

PLOS ONE